# Recent Advances in Eco-Friendly and Scaling-Up Bioproduction of Prodigiosin and Its Potential Applications in Agriculture

**Thi Hanh Nguyen** [1,2], **San-Lang Wang** [2,*] and **Van Bon Nguyen** [3,*]

1 Doctoral Program in Applied Sciences, Tamkang University, New Taipei City 25137, Taiwan
2 Department of Chemistry, Tamkang University, New Taipei City 25137, Taiwan
3 Institute of Biotechnology and Environment, Tay Nguyen University, Buon Ma Thuot 630000, Vietnam
* Correspondence: sabulo@mail.tku.edu.tw (S.-L.W.); nvbon@ttn.edu.vn (V.B.N.)

**Abstract:** Prodigiosin is a red pigment produced by various microbial strains, of these, *Serratia marcescens* has been recorded as the major PG-producing strain. This microbial pigment has attracted much attention because it possesses potential applications in various fields. Thus, this active metabolite has been increasingly studied for bioproduction and investigated for its novel applications. Recently, several prodigiosin-related reviews were reported. These studies covered some aspects of the general physicochemical properties, pathway synthesis, production, and applications of prodigiosin in medicine. However, only a few works discussed the eco-friendly production of prodigiosin from organic wastes. Remarkably, the scaling-up of prodigiosin production and its potential applications in agriculture have rarely been reviewed or discussed. This review extensively presents and discusses the green biosynthesis, enhancement, and scaling-up of prodigiosin production from a wide range of organic byproducts/wastes using different methods of fermentation. Notably, this paper highlights the novel and promising applications of prodigiosin in agriculture via in vitro tests, greenhouse tests, and field studies. The action mechanisms related to some potential bioactivities and the toxicology studies of prodigiosin are also discussed. This review thus supplies scientific data for further research and the application of prodigiosin in the future.

**Keywords:** red pigment; *Serratia marcescens*; bioproduction; agricultural applications

## 1. Introduction

Prodigiosin (PG), a red pigment, is a ring compound belonging to the group of prodiginine with a pyrrolyl pyrromethane skeleton [1]. The structure and physicochemical properties of PG are presented in Figure 1 [2,3]. PG is biosynthesized from various microbial strains, among which, *Serratia marcescens* is recognized as the major producer of PG [4]. This microbial pigment has been extensively investigated and has evidenced potential for applications in many fields, such as medicine, agriculture, industry, food, and the environment [4–22]. Furthermore, its safety was also proved in some works [23–28].

Due to the various benefits of this pigment, the number of original works concerning PG production and the evaluation of its novel uses has increased dramatically in recent years [2,29–34]. To date, various review works on PG have also been published. These studies focused on the topics of general physical–chemical characteristics and biosynthesis pathways of PG [3,35,36], high-level PG production, and the applications of PG in medicine [3,4,35,37,38]. However, the scaling-up of production, action mechanisms, and potential use of PG in agriculture have been rarely mentioned in previous studies. Therefore, this study aims to address these aspects of PG.

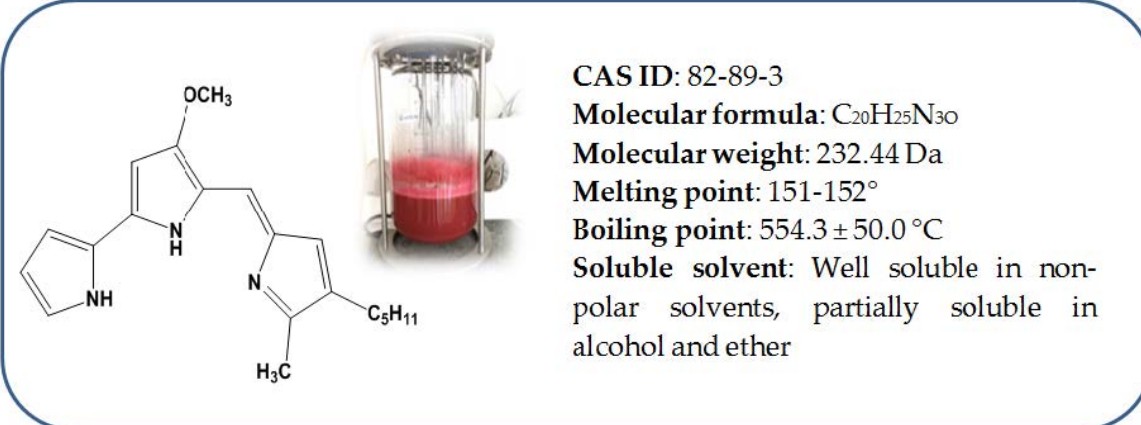

**Figure 1.** The structure and physicochemical properties of prodigiosin.

## 2. Production of Prodigiosin via Microbial Fermentation

Natural bioactive pigments are derived from various sources, such as microbes, insects, and plants [39,40]. Of these, the microbial source possesses the advantage of obtaining bioactive pigment compounds [39]. Obtaining pigments and other secondary metabolites from microbial fermentation is of great interest due to the valuable characteristics of microbes (e.g., fast growth and bioproduction of various secondary metabolites by regulating fermentation conditions) and its green, cost-effective production process (e.g., using an abundant and low-cost substrate, fermentation in mild conditions, and no environmental pollution) [41–47]. Pigments can be produced at high levels by microbial fermentation using low-cost substrates [42–45], resulting in a high capacity for commercial applications [42]. The objective of reducing the cost of pigment production by using inexpensive media is to raise microbial productivity, causing a decrease in the cost of production [43–45]. Subsequently, byproducts/wastes, including agro-industrial byproducts [45], food and kitchen wastes [48,49], and biomass wastes [50], became promising substrate sources for fermentation to produce natural pigments.

PG is also produced via microbial conversion using various kinds of substrates as the carbon/nitrogen source, with the application of scaled-up fermentation to achieve higher productivity in a reduced fermentation time. In this section, different types of substrate sources are discussed to clarify the role of low-cost organic sources and byproducts/wastes compared to the traditional commercial media, with a special focus on the effect of byproduct/waste sources on PG productivity.

### 2.1. Overview of the Substrate Sources for Prodigiosin Production in Liquid-State Fermentation

Liquid-state fermentation (LSF), also known as submerged fermentation, is characterized by the growth of a microorganism in a liquid with high water content and nutrient ingredients that are supplied into a liquid medium during the culture process. The culture parameters and control growth conditions can be easily managed in this fermentation method. The nutrient ingredients and microorganisms are evenly distributed in the medium [51,52]. In fact, LSF is used very commonly in PG production, and many studies used LSF to produce pigment compounds. The production of PG from *Serratia marcescens* by the LSF method using various kinds of substrates as the C/N source is summarized in Table 1.

**Table 1.** The substrate sources for prodigiosin production by liquid-state fermentation.

| Bacteria | Main Substrate | Yield (mg/L$^{-1}$) | Reference |
|---|---|---|---|
| | Commercial media | | |
| *S.marcescens* SMΔR | Luria–Bertani broth | 50 | [53] |
| *S. marcescens* SS-1 | Luria–Bertani broth | 32 | [54] |
| | Yeast extract | 690 | |
| | Yeast extract, axit aspartic | 1400 | |
| | Yeast extract, histidine | 1400 | |
| | Yeast extract, proline | 2500 | |
| *S. marcescens* | Tryptone, glycerol | 123 | [55] |
| *S. marcescens* FZSF02 | Soya peptone | 1774 | [56] |
| | Beef extract | 1699 | |
| | Tryptone | 353 | |
| | Yeast extract | 380 | |
| *S. marcescens* | Nutrient broth | 15 | [57] |
| | Peptone glycerol | 12 | |
| | Tryptone soy broth | 11 | |
| | Luria–Bertani broth | 10 | |
| | Glycerol beef broth | 8 | |
| *S. marcescens* | Nutrient broth | 510 | [58] |
| | Peptone glycerol broth | 300 | |
| | Nutrient broth, maltose | 1820 | |
| *Pseudomonas putida* | Luria–Bertani broth | 17 | [59] |
| | Terrific broth | 94 | |
| *S. marcescens* JNB 5-1 | Luria–Bertani broth | 5830 | [60] |
| | Low-cost organic materials | | |
| *S.marcescens* SMΔR | 4% soybean oil, Luria–Bertani broth | 525 | [53] |
| | 4% olive oil, Luria–Bertani broth | 579 | |
| | 6% sunflower oil, Luria–Bertani broth | 790 | |
| *S. marcescens* FZSF02 | Peanut powder | 3762 | [56] |
| | Peanut powder, soya peptone | 1588 | |
| | Peanut powder, beef extract | 5062 | |
| | Fish meal | 0 | |
| | Soybean powder | 0 | |
| | Corn steep liquor | 0 | |
| | Peanut powder, olive oil, beef extract | 13,622 | |
| *S. marcescens* | Peanut seed broth | 37,600 | [58] |
| | Sesame seed broth | 16,500 | |
| *S. marcescens* | Peanut seed broth | 38,750 | [61] |
| | Peanut oil broth | 2890 | |
| | Sesame seed broth | 16,680 | |
| | Sesame oil broth | 767 | |
| | Copra seed broth | 1940 | |
| | Coconut oil broth | 1420 | |
| *S. marcescens* | Peanut powder, defatted soybean flour | 1282 | [62] |
| *S. marcescens* BWL1001 | Soybean oil, peptone (100/10) | 27,650 | [63] |
| *Streptomyces* fusant NRCF69 | Peanut seed broth | 42,030 | [64] |
| | Sunflower oil broth | 40,110 | |
| | Organic byproduct/waste source | | |
| | Fishery processing byproducts | | |
| *S. marcescens* TNU02 | Demineralized crab shell powder | 4514 | [2] |

**Table 1.** *Cont.*

| Bacteria | Main Substrate | Yield (mg/L$^{-1}$) | Reference |
|---|---|---|---|
| *S. marcescens* TNU01 | Squid pen powder | 3790 | [29] |
| *S. marcescens* TNU01 | Demineralized shrimp shell powder | 5910 | [30] |
| *S. marcescens* CC17 | Shrimp head powder | 5355 | [31] |
| *S. marcescens* TKU011 | Squid pen powder | 978 | [65] |
| | Agro-industrial byproducts | | |
| *S. marcescens* TNU01 | Cassava wastewater | 5202 | [32] |
| *S. marcescens* TUN02 | Peanut oil cake | 5380 | [33] |
| *S. marcescens* UCP 1549 | Cassava wastewater | 49,500 | [66] |
| *S. marcescens* | Peanut oil cake | 40.9 | [67] |
| *S. marcescens* CF-53 | Peanut oil cake | 39,800 | [68] |
| *S. marcescens* | Soybean meal | 9632 | [69] |
| *S. marcescens* UCP 1549 | Corn bran | 1680 | [70] |
| *S. marcescens* MO-1 | Ram horn peptone | 277.74 | [71] |
| *S. marcescens* UTM1 | Brown sugar | 8109 | [72] |
| *Streptomyces* fusant NRCF69 | Dairy processing wastewater broth, Dairy processing wastewater broth, 0.5% mannitol | 36,700 47,000 | [64] |
| | Other byproducts/wastes | | |
| *S. marcescens* UCCM 00009 | Feather and waste frying oil | 9660 | [34] |
| *S. marcescens* | Food waste, Rice husk | 7890 | [73] |
| *S. marcescens* NPLR1 | Tannery solid waste fleshing | 33,000 | [74] |
| *S. marcescens* | Kitchen waste, peptone, proline | 890 | [75] |

### 2.1.1. The Commercial and Low-Cost Organic Materials for Prodigiosin Production

As shown in Table 1, many researchers used commercial media, the most popular being Luria–Bertani broth, yeast extract, beef extract, and nutrient broth, for fermentation to biosynthesize PG with a PG yield of up to 5830 mg/L$^{-1}$ [53–60]. For lower-cost PG production, some studies used inexpensive organic materials for fermentation [4,24,53,56,58,61,63,64]. The medium supplemented with organic materials, such as 4% soybean oil, 4% olive oil, and 6% sunflower oil, recorded an improvement in PG yield of around 525 mg/L$^{-1}$, 579 mg/L$^{-1}$, and 790 mg/L$^{-1}$, respectively, compared to the control commercial medium (50 mg/L$^{-1}$) [53]. Another report explored effective fermentation in media containing various low-cost substrates, and the highest PG yield (13,622 mg/L$^{-1}$) was collected from a medium with a final formulation containing peanut powder, olive oil, and beef extract. Organic materials such as fish meal, soybean powder, or corn steep liquor were not suitable for PG production [56]. Some seeds, seed oils, and coconut oil were also used for PG biosynthesis [58]. The PG yield was as high as 16,680 mg/L$^{-1}$ when using sesame seed broth as substrate and 38,750 mg/L$^{-1}$ when using the peanut seed broth medium. For the other media, PG productivity ranged from 767 to 2890 mg/L$^{-1}$ [58]. The potential PG yield (from 16,500–42,030 mg/L$^{-1}$) was also obtained from media using a substrate, such as seed broth or seed oil [58,63,64]. Thus, seed and seed oil are suitable materials for cost-effective PG production. Overall, low-cost organic sources provide potentially effective PG biosynthesis and reduce the cost of production with improved PG content compared to commercial substrate sources.

2.1.2. The Organic Byproduct/Waste Sources for Prodigiosin Production

To reduce the production cost, some studies used various organic byproducts/wastes (OBWs), including fishery processing byproducts, agro-industrial byproducts, and other organic byproducts/wastes, as C/N sources for microbial production of PG (Table 1).

- Fishery Processing Byproducts

Management of fishery discards, including fishery by-catch and processing byproducts, has been recognized as an emerging research topic [4]. According to data from the Food and Agriculture Organization, around 60% of fishery catch yield is processed, whereas the remaining amount has become a global environmental issue [76]. To resolve this problem, some chitin-rich fishery processing byproducts, such as squid pens, shrimp heads, shrimp, and crab shells, were used to produce high-value microbial secondary metabolites, such as PG (Table 1). Wang et al. [65] used 1.5% squid pens as the sole C/N source for microbial production of PG and obtained a yield of 978 mg/L$^{-1}$ within 2 days of fermentation. Other studies also used fishery discards as the main substrate and showed effective PG production with a productivity of over 3000 mg/L$^{-1}$ [2,29–31]. Squid pen powder (SPP) was the most suitable substrate for PG production by *Serratia marcescens* TNU01 with a yield of 3790 mg/L$^{-1}$ in an optimal medium that contained 1.75% SPP [29]. In fact, squid pens contain chitin/protein in the ratio of 40/60, which might be important for PG biosynthesis by *Serratia marcescens* [24]. In other studies, the culture media were established for *Serratia marcescens* TNU02 fermentation using demineralized crab shell powder (de-CSP) [2]. The suitable formulation for high PG productivity (4514 mg/L$^{-1}$) contained a protein/de-CSP ratio in the range of 3/7 to 4/6 [2]. Two other works also used the byproducts of shrimp processing for fermentation [30,31]. Demineralized shrimp shell powder [30] and shrimp head powder [31] were potential marine byproduct sources for PG production, with high PG yields of 5910 mg/L$^{-1}$ and 5355 mg/L$^{-1}$, respectively. Overall, fishery processing byproducts hold a promising potential when used as the main C/N source for PG production.

- Agro-Industrial Byproducts

Production of natural microbial pigments via fermentation technology is an attractive research target that is featured by high productivity and low cost [77]. Agro-industrial byproducts (AIBs) serve as an ideal substrate for microbial pigment production [77]. AIBs are various residues from the agricultural and food industries [39,78]. Reusing these residues is considered necessary because these sources, if untreated, are estimated to create waste of up to 3.40 billion metric tons by 2050 [50,79]. Thus, various materials from AIBs are also used as C/N sources for fermentation to produce the red pigment PG (Table 1).

Cassava wastewater and peanut oil cake (POC) were considered as two potential AIBs for PG production, and their effectiveness was proven in several reports. Cassava is one of six vital agricultural products worldwide, and the cassava processing industry is also very developed [32,80]. The processing of 1 ton of cassava releases around 300 L of cassava waste [80], which also contains cassava wastewater (CWW). This wastewater source is rich in organic components and potentially causes waste and pollution issues if it is directly released into the environment [81]. Thus, reusing CWW has attracted much interest. Considering that this source contains nutrients that are suitable for microbial growth, some studies utilized CWW as the substrate for fermentation to produce PG [32,66]. Moreover, de Araújo et al. [66] reported that *S. marcescens* UCP1459 fermentation obtained a high PG yield of up to 49,500 mg/L$^{-1}$ in a medium using 6% CWW and 2% mannitol. Recently, Tran et al. [32] also recorded a potential PG productivity of 5202 mg/L$^{-1}$ by utilizing CWW as the main substrate for the culture. POC is also a byproduct collected after oil pressing, and it accounts for over 50% of the substance compared to the original material [33]. This POC byproduct contains very rich nutrients that are suitable for fermentation. POC was also reused as a C/N source for PG production in some works [33,67,68]. According to Vijayalakshmi et al. [67], POC is an inexpensive source, better than other commercial materials for PG production, that could obtain a maximum PG yield of 40.9 mg/L$^{-1}$ in a

5% POC medium. In another report by Naik et al. [68], PG was produced using different agro-wastes, including POC, safflower seed oil cake, cotton seed oil cake, coconut oil cake, and sesame oil cake. When used in agar medium, POC was confirmed to be the most suitable substrate for PG biosynthesis. Furthermore, the PG yield in liquid medium with 8% POC was found to be very promising, achieving up to 39,800 mg/L$^{-1}$ productivity of PG [68]. The latest work by Nguyen et al. [33] also demonstrated the potential of POC in PG production. Notably, after the optimization of culture ingredients, the final medium was designed to be very simple, containing 1% POC with no other commercial supplements, while still capable of producing a high PG content of 5380 mg/L$^{-1}$ [33]. Hence, POC may be a promising, low-cost C/N source with a simple culture formulation for PG production.

Some other sources also showed effectiveness for PG production. The soybean meal medium enhanced the PG yield by 5.19-fold compared to the commercial control medium, achieving an optimal yield of 9632 mg/L$^{-1}$ [69]. PG was produced by *Serratia marcescens* UCP 1549 using corn bran as an inexpensive substrate for fermentation and achieved a PG yield of 1680 mg/L$^{-1}$ [70]. Earlier, ram horns (a byproduct from the meat industry) were also utilized for cost-effective PG production, and it was observed that this substrate promoted both bacteria growth and PG yield (277.74 mg/L$^{-1}$) [71]. Brown sugar was also found to be an inexpensive substrate for PG biosynthesis by *Serratia marcescens* UTM1, and the optimal PG yield was up to 8109 mg/L$^{-1}$ [72]. Some studies used raw materials and wastewater for PG production by *Streptomyces fusant* NRCF69 and found that dairy processing wastewater broth used singly could yield 36,700 mg/L$^{-1}$ of PG. In addition, the pigment content reached up to 47,000 mg/L$^{-1}$ when supplemented with 0.5% mannitol [64].

- Other Organic Byproducts/Wastes for Prodigiosin Production

The low-cost and eco-friendly targets for PG production were also examined using other organic byproducts/wastes. Recently, Atim et al. [34] utilized feather and waste frying oil for PG production and achieved a high PG yield of 9660 mg/L$^{-1}$. Some food waste and fibers are also utilized for PG production [73]. Various fibers, including rice husk, coconut fiber, sawdust, palm oil fiber, sugar cane bagasse, and tender plantain waste, were added to the solid medium used for fermentation. For example, rice husk combined with food waste creates an ideal solid medium for PG biosynthesis that can achieve a PG yield of 7890 mg/L$^{-1}$ [73]. Tannery solid waste fleshing was also used as the main nutrient for PG biosynthesis, which showed a potential productivity of up to 33,000 mg/L$^{-1}$ using a medium containing 3% substrate [74]. Xia et al. [75] used kitchen waste as a nutrient for PG production and achieved a yield of 223 mg/L$^{-1}$ in a medium containing kitchen waste supplemented with 1% peptone and 0.2% proline.

LSF utilizes diverse substrates for fermentation. Considering that many commercial materials are not cost-effective and have low PG productivity, several organic sources have been considered as potential substrates for PG production with high yields in the range of 525–42,030 mg/L$^{-1}$. Notably, although byproducts/wastes are disposal sources, they are surprisingly effective with outstanding pigment productivity, some of which achieve yields as high as 49,500 mg/L$^{-1}$ of pigments. In a review, Han et al. [37] reported that PG production using a low-cost substrate could significantly enhance PG yield by around 1.35–33.7-fold. These results also led to novel research that developed the value of inexpensive materials in the microbial production of potentially bioactive compounds. In addition to advancing economic value, the utilization of byproducts/wastes also helps to avoid the surplus of these nutrient-rich sources and alleviates environmental burdens.

### 2.2. Overview of the Substrate Sources for Prodigiosin Production in Solid-State Fermentation

PG production using the LSF method is popular in research. However, solid-state fermentation (SSF) has also been conducted and assessed for PG production in several investigations (Table 2) [51,82–86]. SSF was conducted on a solid and insoluble substrate with very low moisture (lower limit of 12%), and microorganisms adhered to the solid substrate [87]. Normally, the substrates used in SSF are organic and inexpensive materials which have several advantages including cost-effective fermentation, excellent productivity,

and simple technology [87–89]. Some reports on PG production using SSF show its potential for PG productivity [51,82–86]. Recently, Nguyen et al. [90] used peanut powder for SSF in PG production and reported that the mutant strain *S. marcescens* EMS5 supported up to a 1.52-fold improvement in PG yield compared to the wild strain. Khalid et al. [51] screened some substrates for cultivation and found that wheat bran medium obtained the highest PG yield of 47.5 mg/g$^{-1}$ during early fermentation. Furthermore, a maximal yield of PG (240 mg/g$^{-1}$) was achieved in an optimal medium containing wheat bran supplemented with 0.5 mL/g$^{-1}$ of sunflower oil and 0.4 mL/g$^{-1}$ of live *Bacillus subtilis* cells [51]. Tannery fleshing waste was also used as a solid medium for PG production with a high yield of 70.4 mg/g$^{-1}$ [84]. Another study used a mannitol solid medium and found it to be a promising culture medium for microbial fermentation and PG, yielding a productivity of 3.2 mg/g$^{-1}$ [85]. Some agro-industrial wastes were also utilized as solid mediums for PG biosynthesis [86]. Therein, wheat bran exhibited a maximum PG productivity of 1.307 mg/L$^{-1}$, whereas the other assessed media yielded PG in a lower range from 0.008–0.066 mg/L$^{-1}$ [86]. Other reports attempted to use SSF for PG production using agro-industry wastes such as wheat bran, sugarcane bagasse, instant noodle waste, tangerine peels, pineapple peels, and pineapple crown for fermentation [82]. Among those agro-industry wastes, wheat bran medium showed the highest PG yield. The PG productivity in the optimal medium (5 g wheat bran and 5% waste soybean oil) was recorded at 119.8 g/kg$^{-1}$ dry substrate [82]. PG was also produced by SSF using byproducts, including bagasse, wheat straw powder, and wood chips, as a medium [83]. They recorded bagasse to be the best solid medium for PG production with a yield of 20.13 g/kg$^{-1}$ dry solid, with the PG productivity in optimal medium reaching up to 40.86 g/kg$^{-1}$ dry solid (1.17 g/g$^{-1}$ bagasse, glycerol, and 0.33 g/g$^{-1}$ bagasse soy peptone) [83].

Both the LSF and SSF methods are very commonly used to produce secondary metabolites from microorganisms. These fermentation methods have some advantages and disadvantages [91]. SSF showed high potential for productivity and low production cost; however, the parameters and process of fermentation were rather difficult to control [92]. These issues have limited the use of SSF on a large scale. Nowadays, productivity can be improved using the LSF method by supplementing some yield-enhancing factors during cultivation and reducing the cost, which can be achieved by using inexpensive substrates or byproducts/wastes for fermentation. Notably, fermentation by the LSF method may be performed and scaled up via bioreactor systems to obtain a high PG yield within a significantly reduced cultivation time [2,29–33]. In the future, more research is needed to identify other novel materials that can be used as substrates for fermentation. To date, only fishery processing byproducts have been used in LSF. Thus, there is a need to expand these findings by using other nutrient-rich byproducts that are available in abundant amounts, such as those obtained from food or agro-processing.

**Table 2.** A summary of the substrate sources for prodigiosin production in solid-state fermentation.

| Bacteria | Main Substrate | Yield (mg/g$^{-1}$) | Reference |
|---|---|---|---|
| *S. marcescens* | Ground corn | 3.4 | [51] |
| | Wheat bran | 47.5 | |
| | Rice husk | 11.7 | |
| | Soya bean ground | 10.2 | |
| | Wheat bran, sunflower oil, live cells of *Bacillus subtilis* | 240 | |
| *S. marcescens* UCP 1549 | Soybean oil waste, wheat bran | 119.8 | [82] |
| | Wheat bran | 119.8 | |
| | Sugarcane bagasse | 1.8 | |
| | Instant noodle waste | 66.2 | |
| | Tangerine peels | 22.1 | |
| | Pineapple peels | 18.57 | |
| | Pineapple crown | 31.47 | |

**Table 2.** *Cont.*

| Bacteria | Main Substrate | Yield (mg/g$^{-1}$) | Reference |
|---|---|---|---|
| *S. marcescens Xd-1* | Bagasse | 20.13 | [83] |
| | Wood powder | 15 | |
| | Wheat straw powder | 10.6 | |
| | Bagasse, glycerol, bagasse soy peptone | 40.86 | |
| *S. marcescens* | Tannery fleshing waste | 70.4 | [84] |
| *S. marcescens* UCP/WFCC1549 | Mannitol | 3.2 | [85] |
| *S. matodiphilia* NCIM 5606 | Wheat bran | 1.307 mg/L$^{-1}$ | [86] |
| | Sweet lemon peel | 0.169 mg/L$^{-1}$ | |
| | Orange peel | 0.149 mg/L$^{-1}$ | |
| | Pigeon pea peel | 0.008 mg/L$^{-1}$ | |
| | Rice bran | 0.066 mg/L$^{-1}$ | |
| *S. marcescens* wild | Peanut powder | 858 | [90] |
| *S. marcescens* EMS5 | | 1304 | |

### 2.3. Enhancement of Prodigiosin Production by Supplementing with Nutrients

- Amino Acids

The synthesis of precursor 4-methoxy-2,2-bipyrrole-5-carbaldehyde (MBC) of PG requires the presence of some amino acids, such as proline, serine, and methionine, in the transformation pathway [93,94]. Thus, supplementing these amino acids may promote PG production. Qadri et al. [95] studied the role of methionine in PG biosynthesis via the methylation of pigment. They also found that the addition of a single methionine did not cause pigment production. This process was stimulated for the supplementation of methionine with other amino acids. However, methionine helped to shorten the lag period in the PG biosynthesis process leading to an increase in its productivity [95]. Williams et al. [96] reported a positive impact of amino acids on the improvement of PG production using four amino acids of casein hydrolysate, including l-glutamic acid, l-proline, dl-aspartic acid, and l-alanine. When supplied individually, the amino acids were found to facilitate the synthesis of pigment [96]. Faraag et al. [97] found that serine supplementation could inhibit PG synthesis, whereas tyrosine showed better stimulation than alanine and proline for PG production. The effect of methionine, leucine, proline, alanine, and their combination on PG production was assessed by Siva et al. [98]. They reported that the combination of proline and methionine supported the highest PG yield, which was a 3-fold increase in PG yield compared to the control group.

- Fatty Acids

Another precursor of PG, 2-methyl-3-n-amylpyrrole, is mainly biosynthesized by the oxidation of fatty acids [94]. The addition of some fatty acids from oils could improve PG productivity. According to the results of Chenqiang et al. [56], supplementation with 10 mL/L$^{-1}$ olive oil can enhance PG yield by 9.3-fold compared to the initial medium. In addition, this was the first study to note that 60% PG in pellet form is easier to extract. Giri et al. [61] studied fatty acids in seeds and oils and their effects on improving PG productivity. Their results showed that peanut medium led to a 40-fold increase in PG yield, and three oils (sesame oil, peanut oil, and coconut oil) also improved PG yield, as compared to the commercial medium. The effect of some crude fatty acid sources from groundnut, peanut, sesame, castor, and sunflower seeds on PG yield was explored by Picha et al. [99]. In their tests, the highest PG yield was achieved in media containing crude fatty acid sources from peanuts or sunflowers. A report by Parani et al. [100] noted higher PG productivity when the commercial medium was supplemented with the 4% vegetative oil mixture (coconut, sunflower, and olive oil). Another study by Wei et al. [53], on the effect of the oil-supplement strategy for enhancing PG yield, found that PG productivity

improved with oil supplements used at 2–6% concentrations. Moreover, sunflower oil increased the PG content up to 14-fold in comparison to LB broth [53].

- Microbial Cells

In nature, microbes exist in a complex population, and their interactions are also a factor to be considered for enhancing the production of secondary metabolites [101–103]. Some pigment-producing bacterial strains may increase PG production through their interactions with the living/dead cells of other microbes. PG is anti-bacterial; thus, it can protect the host bacteria from other symbiotic bacteria in culture media. The dead cells of those symbiotic bacteria do not consume nutrients, and so also have no effect on the pigment-producing strain. According to Khalid et al. [51], in a medium with live cells of *Escherichia coli*, *B. subtili*, or *Saccharomyces cerevisiae*, there was a 1.8–2.3-fold increase in PG yield compared to the control medium. Moreover, the heat-killed cells of these microbes also achieved a 1.1–2.1-fold increase in PG yield. Mahmoud et al. [104] reported enhanced PG yield after adding the following bacteria to culture media: *E. coli* cells achieved a 3.8–9.1-fold increase in PG yield, *B. subtilis* cells achieved a 6.7–7 fold increase in PG yield, and *S. cerevisiae* cells achieved about a 2.2–9-fold increase in PG yield. Huy et al. [105] used dead cells of *Lactobacillus rhamnosu* as substrate at different temperatures (70 °C and 100 °C) for PG production. They found that adding 50 μL of heat-killed *L. rhamnosu* cells at 70 °C harvested the best pigment content of 9.79 mg/L$^{-1}$, which was enhanced by 6-fold compared to the system without *L. rhamnosu*. However, killing the *L. rhamnosu* cells at 100 °C tended to reduce pigment production [105]. The mechanisms of these impacts are still unclear. It is possible that the addition of microbes led to the release of some compounds that are effective in enhancing PG production but that have not been identified yet. Alternatively, the direct contact between the PG-producing strain and additional strains could have stimulated PG biosynthesis [51]. One of the reasons for the enhanced PG production associated with the addition of bacterial cells might be related to the adsorption ability, as reported by Wang et al. [106]. They assessed the effect of four *Lactobacilli* strains (TKU 010, TKU 012, BCRC 12193, and BCRC 14011) on the ability of *Serratia marcescens* TKU011 to produce PG in media containing squid pen powder. Among them, *Lactobacillus paracasei* TKU 012 cells showed the highest adsorption rate of up to 84% for the hydrophobic pigment. The PG yield also increased proportionately with the concentration of added TKU 012 cells.

- Mineral Salts

Mineral salts also affect PG productivity. Sulfate and phosphate salts are popularly used in PG-producing media. In a study that used a factorial design to examine the role of four mineral salts in PG biosynthesis by *Serratia Marcescens* BS303, the results showed that NaCl decreased PG content to nearly 20%, whereas a mixture of ammonium iron (III) citrate and anhydrous copper showed the highest PG yield, which was enhanced by 1.8-fold [107]. Suryawanshi et al. [23] explored the effect of some inorganic salts on PG production. Changes in the concentration of NaCl and $CaCl_2$ did not affect PG content, but there was evidence that these salts possibly caused some inhibition of PG biosynthesis at concentrations greater than 2 g/L$^{-1}$. This study also found the following: (1) $CaCO_3$ salt did not change the PG yield at concentrations of 10–100 mg/L$^{-1}$; (2) $KH_2PO_4$ salt showed a maximum PG yield at 80 mg/L$^{-1}$ but reduced PG production if the concentration continued to increase; (3) $FeSO_4$ salt produced a stable amount of PG at 30–50 mg/L$^{-1}$, but the production was significantly reduced at concentrations over 50 mg/L$^{-1}$; and (4) $MgSO_4$ salt at 10–200 mg/L$^{-1}$ led to an increase in PG productivity [23]. Iron is an essential factor for PG production but not for the growth of bacteria [108]. Allen et al. [109] confirmed that the concentration of $Fe^{2+}$ at 0.14 mmol was the most suitable for developing and producing PG by *Vibrio gazogenes* fermentation. Several studies by Nguyen et al. [2,24,29–32] showed a 1.3-fold higher yield of PG in media supplemented with various types of sulfate and phosphate salts, and the optimal concentration was as little as 0.002–0.05% for sulfate and 0.025–0.1% for phosphate. According to these results, some new formulations of sulfate

and phosphate salts were recorded for enhancing PG yield [2,24,30–32]. Nguyen et al. also determined the different optimal formulations of salt using the same bacterial strain (Table 3). Disparities in the optimal salt formulations are possibly due to the difference in nutrient composition of the main substrate used. Research using POC as the main C/N source showed that the supplementation of other ingredients is not necessary when the POC substrate contains enough essential nutrients for fermentation [24].

However, some stimulative salts can inhibit PG production when present in excessive concentrations. Silverman et al. [108] reported that PG production using both the condensation reaction and synthesis reaction of MBC was sensitive to NaCl and that high concentrations of NaCl (3%) inhibited PG biosynthesis. The same result was observed when $Na_2SO_4$, KCl, and $K_2SO_4$ were used at the same concentration. In some reports, the use of phosphate at high concentrations also reduced PG productivity. Witney et al. [110] found that phosphate concentrations between 10 and 250 mM significantly reduced PG yield. Allen et al. [109] reported a similar decrease in PG yield in the presence of $KH_2PO_4$ at concentrations greater than 0.4 mM.

- Other Factors

The supplementation of 5% mineral oil enhanced pigment content by 1.12-fold, as observed by Ulises et al. [62]. The role of α-chitin from shrimp shells and β-chitin from squid pens for PG biosynthesis was reported by Nguyen et al. [24]. They found that α-chitin provided the greatest PG yield (3230 mg/$L^{-1}$), which was higher than the yield produced when supplemented with β-chitin (2730 mg/$L^{-1}$) and control squid pens (2450 mg/$L^{-1}$) [24]. The addition of several sugars into the medium also affects PG production. Lactose supplementation enhanced PG productivity by 18.67% [57]. In another report, sucrose was found to increase PG yield from 8.567 g/$L^{-1}$ to 9.632 g/$L^{-1}$ [69]. According to Giri et al. [61], the addition of maltose or glucose to nutrient broth led to only a 2-fold increase in PG yield, whereas no change was observed for the sesame seed medium. However, in some reports, glucose was confirmed as an inhibitory factor for PG biosynthesis [111–113]. These differences in glucose results may depend on the strain and ingredients of the culture medium.

Overall, the supplement factors, such as amino acids, fatty acids, microorganism cells, mineral salts, and other factors, assist in improving PG productivity from 1.1 to 9.2-fold (Table 3). Thus, besides finding suitable substrates for fermentation, the determination of additional factors is also considered an essential strategy for efficient PG production via fermentation.

**Table 3.** Some supplement factors that assist in enhancing prodigiosin yield.

| Supplement Ingredient | PG Yield | | Unit | Enhancing Yield (Fold) | Reference |
|---|---|---|---|---|---|
| | Initial Medium | Modified Medium | | | |
| dL-alanine (5 mg/$L^{-1}$) | | 19.2 | | 1.5 | |
| l-glutamic acid, l-proline, dl-aspartic acid, and l-alanine (5 mg/$L^{-1}$) | 12.7 | 20 | mg/$L^{-1}$ | 1.6 | [96] |
| l-glutamic acid, l-proline, dl-aspartic acid, and l-alanine (10 mg/$L^{-1}$) | | 22.1 | | 1.7 | |
| l-tyrosine (10 mg/$L^{-1}$) | 2.78 | 8.87 | mg/$L^{-1}$ | 3.2 | [97] |
| Soybean oil (10 mg/$L^{-1}$) | | 9283 | | 5.4 | |
| Canola oil (10 mg/$L^{-1}$) | | 7142 | | 4.1 | |
| Olive oil (10 mg/$L^{-1}$) | 1735 | 11,366 | mg/$L^{-1}$ | 6.6 | [56] |
| Maize oil (10 mg/$L^{-1}$) | | 9119 | | 5.3 | |
| Peanut oil (10 mg/$L^{-1}$) | | 9539 | | 5.5 | |
| Tea oil (10 mg/$L^{-1}$) | | 10,348 | | 6.0 | |

**Table 3.** *Cont.*

| Supplement Ingredient | PG Yield | | Unit | Enhancing Yield (Fold) | Reference |
|---|---|---|---|---|---|
| | Initial Medium | Modified Medium | | | |
| Soybean oil (4%) Sunflower oil (6%) Olive oil (4%) | 152 | 525 790 579 | mg/L$^{-1}$ | 3.5 5.2 3.8 | [53] |
| Live cells of *E. coli* (0.4 mL/g$^{-1}$) Live cells of *B. subtili* (0.4 mL/g$^{-1}$) Live cells of *S. cerevisiae* (0.4 mL/g$^{-1}$) | 100.47 | 225 240 170 | mg/g$^{-1}$ | 2.2 2.4 1.7 | [51] |
| Dead cells of *E. coli* (0.4 mL/g$^{-1}$) Dead cells of *B. subtili* (0.4 mL/g$^{-1}$) Dead cells of *S. cerevisiae* (0.4 mL/g$^{-1}$) | 104.47 | 170 203 120 | | 1.6 1.9 1.1 | |
| Dead cells of *L. rhamnosus* | 1.43 | 9.79 | mg/L$^{-1}$ | 6.8 | [105] |
| Live cells of *E. coli* Live cells of *B. subtili* Live cells of *S. cerevisiae* Dead cells of *E. coli* Dead cells of *B. subtili* Dead cells of *S. cerevisiae* | 450 | 2500 600 2800 4100 3500 4100 | mg/L$^{-1}$ | 5.6 1.3 6.2 9.2 7.8 9.1 | [104] |
| 0.05% MgSO$_4$, 0.03% K$_2$HPO$_4$, | 2450 | 2980 | mg/L$^{-1}$ | 1.2 | [29] |
| 0.02% K$_2$SO$_4$, 0.05% K$_2$HPO$_4$ | 3980 | 5200 | mg/L$^{-1}$ | 1.3 | [30] |
| 0.02% K$_2$SO$_4$, 0.025% Ca$_3$(PO$_4$)$_2$ | 3862 | 4500 | mg/L$^{-1}$ | 1.2 | [31] |
| 0.02% (NH$_4$)$_2$SO$_4$, 0.1% K$_2$HPO$_4$ | 3010 | 4000 | mg/L$^{-1}$ | 1.3 | [2] |
| 0.05% MgSO$_4$, 0.1% K$_2$HPO$_4$ | 3981 | 5202 | mg/L$^{-1}$ | 1.3 | [32] |
| 0.05% K$_2$HPO$_4$, 0.1% CaSO$_4$ | 3230 | 4320 | mg/L$^{-1}$ | 1.3 | [24] |
| α-chitin (5/3 *w/w*) | 2450 | 3230 | mg/L$^{-1}$ | 1.3 | [24] |
| β-chitin (2/6 *w/w*) | 2450 | 2730 | mg/L$^{-1}$ | 1.1 | [24] |
| Lactose | 15 | 17.8 | mg/L$^{-1}$ | 1.2 | [57] |
| 0.864% Sucrose | 8.567 | 9.632 | g/L$^{-1}$ | 1.1 | [69] |
| 0.5% Maltose 0.5% Glucose | 520 | 1836 1689 | mg/L$^{-1}$ | 3.5 3.2 | [61] |

### 2.4. Scaling-Up Production of Prodigiosin

Bioreactors are used for the large-scale production of secondary metabolites [114]. In large-scale PG production, the bioreactor system has been used. Some fermentation strategies in bioreactors include batch processes, fed-batch processes, and continuous cultivation [29,115,116]. Among those, batch processes are mainly used for research on PG production [29–33,67,68,72,117–120] because they provide several advantages including a short duration, unlikely contamination as no nutrients are added, and an easily managed culture process [121]. The fed-batch process is a partly open system, also called a semi-continuous bioreactor, characterized by the continuous or intermittent addition of the required nutrients to the initial medium after the start of fermentation or from the mid-period of the cultivation process. The advantage of this process is that it achieves a higher overall product content. Tao et al. [115] designed a two-step culture process, wherein the first stage used glucose as the main carbon source for cell growth, and in the next stage, PG was produced by switching glucose to glycerol in the medium. A 5-L bioreactor yielded a PG production efficiency that was 7.8 times higher (583 mg/L$^{-1}$) than the original cultivation mode (75 mg/L$^{-1}$) with glycerol as the sole carbon source [115]. After a batch growth phase, an equilibrium is established with respect to a particular component

(also called a steady state). The fresh medium is added into the batch system along with a corresponding withdrawal of the medium containing the product at the exponential phase of microbial growth. These bioprocesses are referred to as continuous cultures. Continuous cultivation gives a near-balanced growth, reduces product inhibition, and improves space-time yield [121]. However, the long cultivation period also increases the risk of contamination and may lead to long-term changes in the cultures, including the impact on the product yield. In a study, PG was biosynthesized by continuous fermentation of *Hahella chejuensis* in 10 L and 200 L bioreactors with glucose (2.5 g/L$^{-1}$) as the most suitable carbon source. The 10 L bioreactor achieved 2280 mg/L$^{-1}$ of PG yield in 49 h, whereas the 200 L system only recorded a PF yield of 1305 mg/L$^{-1}$ after 10 days of fermentation [116]. Luis et al. [122] performed a continuous fermentation, and the PG content was 1.2-fold higher compared to that obtained in the batch mode. A continuous fermentation was performed with a 5-fold higher nutrient concentration than that of the initial media to take advantage of a large amount of foam from fermentation. The result was a 2.7-fold increase in PG yield compared to that obtained by batch fermentation [117].

Several bioreactor systems have been examined for PG production. Therein, nearly all research systems have used laboratory-scale bioreactor systems with small and medium capacities from 2.5 to 20 L [29–33,66–68,72,115,118–120,122,123]. On a larger scale, Qi et al. [117] performed an experiment in a 50 L bioreactor using 47.8 L of a medium. PG was also produced on an industrial scale [116], wherein continuous fermentation was carried out in two different fermenters of sizes 10 L and 200 L. Similar to flask fermentation, *S. marcescens* is the most popular and widely used strain for fermentation in a bioreactor [29–33,66–68,72,115,117–119,122,123]. Although, Jeong et al. [116] used *Hahella chejuensis* for PG production.

To achieve a large-scale, lower cost PG production and reduce environmental pollution, some studies utilized byproducts from food processing or agriculture for fermentation [29–33,67,68,72,123]. In addition to several commercial substrate sources popularly used in fermentation, such as nutrient broth, maltose, peptone, casein, sucrose, glycerol, and glucose [115–119,122], PG was also biosynthesized by utilizing inexpensive byproduct/waste [29–33,67,68,72,123,124]. Nguyen et al. utilized some byproducts and wastes, such as squid pens [29], shrimp shells [30], shrimp heads [31], crab shells [2], and agriculture processing byproducts, including POC [33], cassava wastewater [32] for PG production by fermentation. Vijayalakshmi et al. [67] and Naik et al. [68] used POC as a C/N source for PG production. Aruldass et al. [72] designed a medium containing brown sugar, a kind of byproduct from sugar processing. Compared to commercial substrates, the efficiency of creating PG from these low-cost materials was relatively better and sometimes higher. The yield of PG increased from 583 to 13,100 mg/L$^{-1}$ when using commercial medium, whereas using the byproduct substrate for fermentation showed an average PG yield of about 6000 mg/L$^{-1}$ and a maximum of 40,000 mg/L$^{-1}$ (based on the results synthesized in Table 4).

Fermentation in a bioreactor is also efficient in terms of productivity and shorter time compared to flask cultures. Nearly all research on PG production in bioreactors reported a 1–76.7-fold increase in the yield of PG. Nguyen et al. [2,29–33] reported that fermentation in a bioreactor only required a short time to reach the maximum level of PG expression. Normally, a small-scale fermentation (in the flask) takes up to 2 days to reach the maximal yield, whereas a large-scale fermentation (in a bioreactor) requires only 8–10 h.

Overall, when PG is produced on a large scale, efficiency is achieved in terms of both yield and production time. To date, there have been several reports on PG production; however, almost all have been conducted on a small scale, and there are only a few reports on large-scale PG production. Moreover, nearly all studies have used the bioreactor system on a laboratory scale, and more research is still needed to assess the potential for industrial scale PG production. In particular, other kinds of byproducts or waste for fermentation should be screened for better biosynthesis efficiency and cost effectiveness.

**Table 4.** The efficiency of prodigiosin production in large-scale (bioreactor system).

| Strain | Substrate | Max Prodigiosin Yield (mg/L$^{-1}$) in Fermentation Time (hours) | | Culture Volume (L)/Bioreactor (L) | Enhancing Yield (Fold) | Reference |
| --- | --- | --- | --- | --- | --- | --- |
| | | In Flask | Bioreactor | | | |
| *S. marcescens* TUN02 | demineralized crab shell powder, casein, $(NH_4)_2SO_4$, $K_2HPO_4$ | 4514 (36 h) | 5100 (8 h) | 4.5 L (15 L) | 1.1 | [2] |
| *S. marcescens* TNU01 | Squid pens powder, $K_2HPO_4$, $MgSO_4$ | 3790 (48 h) | 3450 (12 h) | 3 L (10 L) | - | [29] |
| *S. marcescens* TUN02 | Demineralized shrimp shell powder, casein, $K_2SO_4$, $K_2HPO_4$ | 5910 (36 h) | 6200 (8 h) | 5 L (15 L) | 1 | [30] |
| *S. marcescens* CC17 | Shrimp head powder, casein, $K_2SO_4$, $Ca_3(PO_4)_2$ | 5355 (60 h) | 6310 (8 h) | 6.75 L (12 L) | 1.2 | [31] |
| *S. marcescens* TNU01 | Cassava wastewater, casein, $MgSO_4$, $K_2HPO_4$ | 5202 (48 h) | 6150 (8 h) | 7 L (14 L) | 1.1 | [32] |
| *S. marcescens* TUN02 | Peanut oil cake | 5380 (48 h) | 6886 (10 h) | 4 L (14 L) | 1.3 | [33] |
| *S. marcescens* | Peanut oil cake | 40.9 (30 h) | 50 (30 h) | 1.5 L (3 L) | 1.2 | [67] |
| *S. marcescens* CF-53 | Peanut oil cake | 39,800 (42 h) | 40,000 (42 h) | 1 L (2 L) | 1 | [68] |
| *S. marcescens* UTM1 | Brown sugar | 237 (24 h) | 8109 (24 h) | 5L (5L) | 34.2 | [72] |
| *S. marcescens* B6 | Two-step feeding strategy with glycerol | ND | 583 (30 h) | 2.5 L (5 L) | ND | [115] |
| *Hahella chejuensis* | Glucose (Continuous fermentation) | 448.1 (24 h) | 2280 (49 h) | 5 L (10 L) | 5 | [116] |
| | | | 1305 (240 h) | 100 L (200 L) | 2.9 | |
| *S. marcescens* NS-17 | Maltose, peptone, Tween-80, soybean oil, NaCl, KCl | 60.5 (56 h) | 4644.6 (56 h) | 47.8 L (50 L) | 76.7 | [117] |
| *Serratia* sp. KH-95 | HP-20 resin, casein, $K_2HPO_4$, $MgSO_4$, NaCl | ND | 13,100 (30 h) | 1 L (2.5 L) | ND | [118] |
| *S. marcescens* | Sucrose, peptone | 391.1 (48 h) | 595 (48 h) | 6.5 L (7 L) | 1.5 | [119] |
| *S. marcescens* ATCC 27117 | Nutrient broth | 13,600 (24 h) | 7800 (20 h) | 3 L (4.5 L) | - | [120] |
| *S. marcescens* BS 303 | Peptone, glycerol, mineral broth, TritonX-114 | 540 (24 h) | 872 (62 h) | 0.935 L (1.5 L) | ND | [122] |
| *Serratia* AM8887 | Fertilizer waste, sucrose, glycerol, NaCl | ND | 7316 (19 h) | 17.8 L (20 L) | ND | [123] |

ND: No determine.

## 3. The Potential Applications of Prodigiosin in Agriculture

In this section, the bioactivities related to the potential applications of PG in agriculture are presented and discussed. Focus is given to the anti-fungal, anti-bacterial, anti-insect, and anti-nematode effects via in vitro tests, in the greenhouse, and in the field.

### 3.1. The Potential Applications in Agriculture: In Vitro Tests

3.1.1. Anti-Fungal Activity of Prodigiosin

Many harmful fungi have been reported on crops [125,126]. The anti-fungal activity of PG makes it a potential candidate for use in the agricultural industry to protect crops from fungal harm (Table 5) [23,74,100,124,127–133].

**Table 5.** Anti-fungal activity of prodigiosin for agriculture.

| Fungi | The Unit of Antifungal Activity | Value | Reference |
|---|---|---|---|
| *Aspergillus flavus*<br>*Fusarium oxysporum* | MIC—Minimum inhibitory concentration (µg/mL) | 10<br>8 | [23] |
| *Aspergillus niger*<br>*Fusarium moniliforme* | | 230<br>210 | [74] |
| *Helminthosporium sativum*<br>*Curvularia lunata*<br>*Alternaria alternate*<br>*Fusarium oxysporum*<br>*Cercospora apii*<br>*Rhizoctonia solani* | Diameter of inhibition zone (mm) | 42<br>40<br>40<br>30<br>24<br>11 | [100] |
| *Alternaria* sp.<br>*Fusarium* sp. | MIC—Minimum inhibitory concentration (µg/mL) | 80<br>160 | [124] |
| *Didymella applanata* | $IC_{50}$—Concentration to inhibit 50% fungal (nmol/mL) | 2.5 | [127] |
| *Phoma lingam*<br>*Sclerotinia sclerotiorum* | Hyphal growth diameter (%) | 25<br>60 | [128] |
| *Botrytis cinerea* | Inhibition of spore germination (%) | 0–80 | [129] |
| *Mycosphaerella fijiensis* | $IC_{50}$—Concentration to inhibit 50% fungal (µg/mL)<br>Inhibits growing germ tubes (%) | 996<br>63 | [130] |
| *Cochliobolus miyabeanus*<br>*Fusarium moniliforme*<br>*F. oxysporum f.* sp. *allii*<br>*F. oxysporum f.* sp. *raphani*<br>*F. oxysporum f.* sp. *cucumerinum*<br>*F. oxysporum f.* sp. *spinaciae*<br>*F. oxysporum f.* sp. *cepae*<br>*F. roseum*<br>*F. solani var. coeruleum*<br>*F. ventricosum*<br>*Phytophthora cactorum*<br>*P. capsici*<br>*P. castaneae*<br>*P. citrophthora*<br>*P. infestans* sp.<br>*P. melonis*<br>*Pyricularia oryzae*<br>*Pythium spinosum*<br>*Pythium ultimum*<br>*Rhizoctonia solani* sp. | Growth inhibition (%) | 83.3<br>5.9<br>17.6<br>16.3<br>23.5<br>1.2<br>17.8<br>5.8<br>14.3<br>11.1<br>89.7<br>63.5<br>74.5<br>85.1<br>80.5–83.2<br>93<br>28.8<br>66.4<br>44.2<br>17.6–52.9 | [131] |
| *Pythium myriotylum*<br>*Rhizoctonia solani*<br>*Sclerotium rolfsii*<br>*Phytophthora infestans*<br>*Fusarium oxysporum* | Growth inhibition (%) | 71.33<br>61.33<br>49.33<br>48.66<br>31 | [132] |
| *Colletotrichum nymphaeae* | Inhibits germination (%) | 100 | [133] |

Most studies focused on the activity of PG in its purified form. Hiroshi et al. [131] reported the inhibitory capacity of PG against 20 species of plant-pathogenic fungi belonging to six genera. At a PG concentration of 10 $\mu g/mL^{-1}$, the growth of all tested fungal species was inhibited, some by more than 80% [131]. Nobutaka et al. [129] observed that PG controlled the growth rate of *Botrytis cinerea*, a gray mold fungus of cyclamen. The spore germination rate was reduced by one-third at 1 $\mu g/mL^{-1}$ of PG compared to the treatment using an enzyme mixture. Duzhak et al. [127] assessed the inhibitory ability of PG against a pathogen fungus of raspberries (*Didymella applanata*) and found that PG was the main factor that suppressed fungal growth at an $IC_{50}$ value of 2.5 $nmol/mL^{-1}$. The role of PG and chitinase from *Serratia marcescens* against the fungus *D. applanata* was also assessed, and they found that PG was a key anti-fungal agent, with an $IC_{50}$ value of 2.5 $nmol/mL$. In contrast, the chitinase production did not provide the bacterium with any competition for the fungus [127]. Suryawanshi et al. [23] studied the potential activity of PG against two pathogenic fungi, including *Aspergillus flavus* and *Fusarium oxysporum*, and observed that their minimum inhibitory concentrations (MIC) were 10 $\mu g/mL^{-1}$ and 8 $\mu g/mL^{-1}$, respectively. Another report studied the inhibitory activity of PG against *A. niger* and *F. moniliforme* and found their MIC values to be 230 $\mu g/mL$ and 210 $\mu g/mL$, respectively [74]. In the work of Martha Ingrid et al. [130], PG showed an efficient inhibitory activity against *Mycosphaerella fijiensis*, a fungus that causes black Sigatoka disease in bananas. At a concentration of 14.3 $\mu g/mL^{-1}$, this pigment inhibited 60% of the growth of germ tubes, and the $IC_{50}$ value was 996 $\mu g/mL^{-1}$ [130]. Sagar et al. [124] tested the bioactivity of PG against two popular fungi, *Alternaria* and *Fusarium*, at MIC values of 80 $\mu g$ and 160 $\mu g$, respectively. The red pigment was also tested by Samer et al. [128] for the inhibition of two types of fungi: the agent of blackleg disease and white mold on crops. They observed that 50 $\mu M$ PG significantly inhibited the growth of *Phoma lingam* with nearly 25% hyphal growth, whereas its effect on *Sclerotinia sclerotiorum* w relatively insignificant [128]. Most recently, in 2022, there was a report of antifungal activity by purified PG from *Serratia rubidaea* Mar61-01 against *Colletotrichum nymphaeae*, the agent that causes anthracnose disease in strawberries. PG showed an ability to inhibit the germination of this fungi by up to 100% [133].

In some studies, the crude extract of the PG pigment was used to evaluate its anti-fungal activities. Parani et al. [100] evaluated the effect of crude PG from *Serratia marcescens* SR1 on the inhibition of some pathogen fungi using the well-diffusion method. Therein, crude PG showed a maximum inhibitory zone (42 nm) against *Helminthosporium sativum* and five other types of fungi including, in decreasing order, *Curvularia lunata* (40 mm), *Alternaria alternate* (40 mm), *Fusarium oxysporum* (30 mm), *Cercospora apii* (24 mm), and *Rhizoctonia solani* (11 mm) [100]. Jimtha et al. [132] evaluated the anti-phytopathogenic activity of crude extract from *S. marcescens* Bm1 and observed significant growth inhibition of *Pythium myriotylum* (71.33%), whereas the inhibitory activity against other fungal strains, such as *Phytophthora infestans*, *Sclerotium rolfsii*, and *Rhizoctonia solani*, was less than 62%. The crude pigment was also found to be effective against *Fusarium oxysporum* (31% inhibition), although the effect was non-significant [132].

PG is an effective, promising candidate against the inhibition of many fungal species. Many strains, including those of *Fusarium* sp. and *Phytophthora* sp., were tested. However, further research is needed to assess the impact of this pigment against other harmful phytopathogens. To date, almost all studies have used purified PG for tests, whereas only a few have used the crude extract to assess the potential activity of PG. Furthermore, the direct treatment by crude extract may possibly have more advantages. Thus, the agricultural bioactivity of the crude pigment should be more exhaustively evaluated.

### 3.1.2. Prodigiosin Inhibits Other Organisms That are Harmful to Crops

In addition to its inhibitory effect against phytopathogenic fungi, PG is also active against other pathogens, such as bacteria, nematodes, and insects, that are harmful to crops. Several studies showing the bioactivity of PG against these pathogens are summarized in Table 6 [33,65,128,131,134–139].

**Table 6.** The bioactivity of prodigiosin against bacteria, nematodes, and insects harmful to crops.

| Object | The Unit of Activity | Value | Reference |
|---|---|---|---|
| Bacteria | | | |
| *Acidovonax avenae* | | 50 | |
| *Agrobacterium tumefaciens* | | 50 | |
| *Clavibacter michiganensis* subsp. *michiganesis* | | 6.3 | |
| *Erwinia carotovora* subsp. *carotovora* | | 25 | |
| *E. herbicola* | | 50 | |
| *Pseudomonas cichorii* | | >100 | |
| *P. fluorecens* | | >100 | |
| *P. gladioli* pv. *gladioli* | | >100 | |
| *P. glumae* | MAC—The maximal allowable concentration ($\mu g/mL^{-1}$) | >100 | [131] |
| *P. mariginalis* pv. *marginalis* | | >100 | |
| *P. syringae* pv. *lachrymans* | | >100 | |
| *P. syringae* pv. *mori* | | >100 | |
| *P. syringae* pv. *phaseolicola* | | >100 | |
| *P. syringae* pv. *pisi* | | >100 | |
| *Ralstonia solanacearum* sp. | | >100 | |
| *Xanthomonas campetris* pv. *campestris* | | 25 | |
| *X. campetris* pv. *carotae* | | 50 | |
| *X. campetris* pv. *oryzae* | | 25.5 | |
| Nematode | | | |
| *Heterodera schachtii* | IC$_{50}$ ($\mu$M) | 13.3 | [128] |
| *Radopholus similis* | IC$_{50}$ ($\mu g/mL^{-1}$) | 83 | [134] |
| *Meloidogyne javanica* | | 79 | |
| *Meloidogyne incognita* | IC$_{50}$—Anti juvenile ($mg/mL^{-1}$) | 0.2 | [33] |
| | IC$_{50}$—Anti egg-hatching ($mg/mL^{-1}$) | 0.32 | |
| | IC$_{50}$—Anti juvenile ($mg/mL^{-1}$) | 31.9 | [135] |
| Insect | | | |
| *Diaphorina citri* | The inhibitory rate of oviposition (%) | 42 | [136] |
| | The moderate inhibitory rate of egg hatch (%) | 26 | |
| *Helicoverpa armigera* | Larval mortality rate (%) | 70–100 | [137] |
| *Spodoptera litura* | | | |
| *Drosophila* | IC$_{50}$—Anti larval (ppm) | 230 | [138] |
| | IC$_{50}$—Anti larval ($g/L^{-1}$) | 0.23 | [65] |
| *Spodoptera litura* | The mortality at 8$\mu$g/g diet (%) | 3 | [139] |
| *Plutella xylostella* | | 96 | |
| *Adoxophyes honmai* | | 12 | |

IC50—Concentration to inhibit 50%.

Hiroshi et al. [131] tested the antibacterial activity of PG against plant pathogenic bacteria belonging to 19 species in six genera. PG was the most active against *Clavibacter michiganensis* subsp. *michiganensis* with the maximal allowable concentration (MAC) of 6.3 $\mu g/mL^{-1}$ [131]. PG was also found to be active against the following bacteria: *Erwinia*

*carotovora* subsp. *carotovora* and *Xanthomonas campetris* pv. *oryzae* with MAC values of 25 μg/mL$^{-1}$ and 25.5 μg/mL$^{-1}$, respectively; and four strains including *Acidovonax avenae*, *Agrobacterium tumefaciens*, *Erwinia herbicola*, and *X. campetris* pv. *carotae* that were inhibited at similar MAC values of 50 μg/mL$^{-1}$. PG displayed a weaker activity against the other tested strains with MAC values greater than 100 μg/mL [131]. Research related to the objective of using PG against plant pathogenic bacteria is still very limited despite the fact that plant bacterial pathogens are spreading widely across the globe [140,141]. PG is also known as a potential inhibitor for many gram-negative and gram-positive bacteria; thus, its potential anti-bacterial role is relatively important.

Plant-damaging nematodes can seriously damage important crops across the world [142]. They can reduce the productivity of crops by 5–10% in developed countries [143] and reduce the quality of agricultural products [144]. Therefore, finding a solution to manage these disease nematodes is necessary [145]. PG has the potential to inhibit some of these crop-damaging nematode species. Samer et al. [128] assessed the impact of the Prodiginine group on the plant parasitic nematode *Heterodera schachtii*; only PG caused considerable paralysis of the second-stage juveniles (J2s) with an IC$_{50}$ value of 13.3 μM, whereas the remaining compounds were less effective against J2s [128]. This pigment was also effective against the juvenile stage of *Meloidogyne javanica* and *Radopholus similis* at low concentrations, with IC$_{50}$ values of 79 and 83 μg/mL$^{-1}$, respectively [134]. At the same tested concentrations, the PG pigment could completely inhibit egg-hatching, and no juveniles were observed [134]. Some reports were conducted to assess the activity of PG against *Meloidogyne incognita*. The culture filtrate of *Serratia marcescens* was used for testing the activity of newly hatched juveniles of *Meloidogyne incognita* after 72 h of treatment [135]. The crude pigment was effective against this nematode even at a low concentration, with an IC$_{50}$ value of 31.9 mg/mL$^{-1}$ [135]. This proved the effectiveness of using the microbial secondary metabolites, in comparison with the use of whole organisms. Recently, Nguyen et al. [33] studied the activity of purified PG for *Meloidogyne incognita* on both the J2s and egg-hatch. They found that the purified compound showed a stronger inhibition compared to the crude pigment, and the IC$_{50}$ was as low as 0.2 mg/mL$^{-1}$. These results showed the novel inhibitory effect of PG against the egg-hatching of this nematode with an IC$_{50}$ value of 0.32 mg/mL$^{-1}$ [33].

Some studies used the inhibition of PG to target harmful plant insects. Wei et al. [136] evaluated the potential activity of PG against Asian Citrus Psyllid. The toxicity of PG against nymphs depended on temperature. Based on the test results, the most suitable temperature was confirmed to be 30 °C, which was used for follow-up experiments. PG was significantly effective against the oviposition and egg-hatching of *Diaphorina citri* with an inhibitory rate of 42%. PG also exhibited a moderate inhibitory rate against egg-hatching at a rate of 26%. Moreover, treatment against adult hoppers with an LC20 and LC50 solution of purified PG at 30 °C was recorded to excrete less honeydew, as compared to the control [136]. This pigment also demonstrated insecticidal effects against *Helicoverpa armigera* and *Spodoptera litura* larvae, with mortality rates of 70 and 100% at 20 and 30 mg/mL$^{-1}$, respectively [137]. The activity of pure PG against *Drosophila* larvae was reported in some studies [65,138]. According to Wang et al. [65], after 5 days of treatment with pure PG, the IC$_{50}$ value was recorded at 0.23 g/L$^{-1}$. Furthermore, to confirm the role of PG against *Drosophila*, the PG concentration in each test formula showed high insecticidal activity, which corresponded to a high PG concentration [65]. In a study by Liang et al. [138], PG presented effective toxicity against *Drosophila* with an IC$_{50}$ value of 230 ppm compared to other commercial food colorants. Asano et al. [139] accessed the anti-insect ability of PG against three kinds of insects, including *Spodoptera litura* (a common cutworm), *Plutella xylostella* (the diamondback moth), and *Adoxophyes honmai* (the tea tortrix). PG showed the highest activity against *Plutella xylostella* with high mortality of up to 96% at an 8 μg/g$^{-1}$ diet [139]. *Spodoptera litura* and *Adoxophyes honmai* were less sensitive to PG, with the mortality only reaching 3% and 12%, respectively [139].

These summarised study results show that the anti-fungal effect of PG was frequently confirmed; however, only a few studies mentioned other bioactivities. Thus, more studies are needed to explore the activities of PG on many other objects. This review enriches the existing information about the future bioactive and potential applications of PG. Recently, there has been a push to develop green and sustainable agriculture, whereby natural biological inhibitory agents are considered effective and safe solutions for plant disease management. PG possesses the ability to inhibit many harmful agents and may become a potential candidate for controlling various plant diseases in the future of agriculture production.

### 3.2. The Potential Applications of Prodigiosin via Studies in Greenhouses and in the Field

The effect of the culture fluid from *Serratia rubidaea* Mar61-01 on the growth of *Colletotrichum nymphaeae* (the causal fungi of strawberry anthracnose) was tested in vivo and in greenhouse conditions [133]. In the in vivo condition, the culture fluid containing pigment significantly decreased fruit decay with a bio-control efficacy of 48.60%. The tested sample also showed a potential effect in the greenhouse test for strawberry anthracnose reduction. The spraying method of culture fluid was highly effective, with a 72.22% decrease in strawberry anthracnose disease compared to the drenching soil method, which only reduced the disease by 44.44% [133]. In a report by Roberts et al. [146], the purified PG from *Serratia rubidaea* was used to determine the damping-off suppression of cucumber caused by *Pythium ultimum* fungus. In the presence of *Pythium ultimum* inoculum, the cucumber plants with seeds that were treated by PG showed a greater development compared to the nontreated plants. In addition, the ethanol extract of mutant strain Tn246 non-producing PG could not control the disease caused by this pathogen. This study helped to clearly understand the role of PG in the management of these phytopathogens [146].

Samer et al. [128] evaluated the bioactivity of prodiginines against the first stage of plant parasitic nematodes (*Heterodera schachtii*) during the growth of a plant using *Arabidopsis thaliana* as the model plant. The infection results revealed that treatment with PG reduced the total number of *H. schachtii* individuals that developed in the *Arabidopsis thaliana* plant by up to 50%. Furthermore, the size of female nematodes and their associated syncytia were smaller after processing. PG also promoted the growth of the plant depending on treatment concentration. Some reports assessed the role of PG against the knot-root nematode *Meloidogyne incognita* in greenhouse conditions [135,147]. Omnia et al. [135] used the culture broth and culture filtrate of *Serratia marcescens* for testing *Meloidogyne incognita* inhibition in vivo on tomato seedlings. Both applications effectively suppressed the nematode population, both in the soil and root, and promoted shoot and root lengths and plant biomass, compared to that in the untreated plants. Similar results were found in the developmental stages, females, and eggs within tomato roots. Nguyen et al. [147] used the fermented product of *Serratia marcescens* TNU02 which is rich in PG content to evaluate the anti-nematode *Meloidogyne incognita* on the black pepper plant model in greenhouse conditions. The fermented product of the TNU02 strain (at the treatment dose of 80 mL) exhibited potent activity against nematodes on pepper root and soil with up to 70% and 85% mortality rates, respectively. It especially reduced the number of nematodes by 4-folds in the root nodules. In addition, the influence of the fermented product on growth targets was determined via growth parameters and chlorophyll content. Among various treatment concentrations, the use of 40 mL and 80 mL of the fermented product was more potent than other treatments at other concentrations. This seems to be a promising application of PG for the management of black pepper nematodes in agriculture. In another study, the purified PG from *Serratia Marcescens* KH-001 was applied to determine whether it could protect orange orchards against Asian Citrus psyllid [136]. The field experiments showed that a 10% PG emulsifiable concentration was highly effective (up to 70–100%) compared to the other concentrations studied. Furthermore, the results recorded in July and August were better than in October [136].

In addition to its potential activity in vitro, PG also showed promising impacts during tests conducted in vitro, in the greenhouse, and in the field. However, few current studies have assessed the bioactivities of this pigment on a larger scale, especially in greenhouse conditions or the field. Thus, further studies are needed on the application of PG in agriculture. To date, the evaluation of PG activity has been mainly reported in the pure compound form, with only a few studies examining its bioactivities at the nanoscale. Furthermore, nano-PG has been mainly used to assess activities in the medical and dye industries [148–155]. However, few studies have assessed PG nanoparticle bioactivities for potential use in agriculture. Currently, green synthesis by nanotechnology combined with natural active ingredients is being explored with the objective to improve the formulations of pesticides in terms of effectiveness, safety, and environmental protection [156–160]. Thus, investigation of the potential uses of PG in nano forms may find new ways to enhance the bioactivity and stability of PG.

### 3.3. Mechanisms of Potential Bioactivities of Prodigiosin

The mechanism of PG for bioactivities in medicine has been demonstrated in many reports [18,161–169]. The anti-cancer activity of PG was studied the most, and some mechanisms were clarified according to each type of cancer [164–169]. PG is known as a potential antibiotic for many gram-negative and gram-positive bacterial strains. The anti-bacterial capacity of prodigiosin is based on its pH modification, inhibition of cell cycles, and bacterial genetic machinery cleavage [18]. Darshan et al. [161] mentioned other mechanisms, including the formation of reactive oxygen species (ROS), whereby PG infiltrates and is localized on the membrane and the nucleus of bacteria cells. The PG then promotes a ROS reaction that damages cells and causes the death of the bacterial cells. In other reports, the anti-bacterial role of PG was proved via an induction process with autolysins, enzymes attending to hydrolysis, and the restructuring of peptidoglycan on bacteria cell walls [162]. PG causes an increase in the permeability of cell walls. It also plays a role in proton transport and ATP synthesis. Therefore, PG has become an ideal factor for the induction of autolysins, causing bacterial death [162]. Moreover, PG causes bacterial inhibition by reducing the respiration of cells and inhibiting protein and RNA synthesis [163]. Yip et al. [18] suggested that PG also inhibits two strategies that harmful bacteria use when they compete with others in the medium. PG can inhibit the growth of bacteria and virulence factors such as hemolysin, protease production, and biofilm formation [18]. The mechanism of PG differs depending on each type of microbe [162,163].

Considering its role in agriculture, the action mechanism of PG is still rarely discussed. Several reports proved the roles of PG against zygomycete and ascomycete fungi [170] and the *Meloidogyne incognita* nematode [33]. A study on the effect of *Serratia marcescens* in the inhibition of zygomycete and ascomycete fungi showed that bacteria could not infiltrate inside cell membranes and cause fungal death. However, the activity of PG increased the permeability of cell membranes, and the *Serratia marcescens* then easily infiltrated inside the fungal hyphae [170]. Some molecular mechanisms of anti-nematode activity were proved by the inhibition of enzyme or protein targets by virtual screening assays [171–175]. Nguyen et al. [33] suggested the anti-nematode role of PG (*Meloidogyne incognita*) via the inhibition of the enzyme acetylcholinesterase (AChE). This enzyme is often used to evaluate anti-nematode ability [172]. Based on the previous report by Nguyen et al. [31], a docking study was conducted to assess the interaction and inhibitory efficacy of the ligand toward the target enzyme. PG showed strong binding energy (−12.3 kcal/mol), and up to six linkages (interactions with amino acids) in the binding sites were found on AChE.

Overall, reports on the molecular mechanism of PG related to its bioactivities for potential application in agriculture are still very limited. Although PG was tested and promising inhibition has been recorded on many organisms harmful to crops, its mechanism of action on each object is still not explored fully. Thus, understanding the mechanism of action is needed.

*3.4. The Toxicology of Prodigiosin*

The safety of PG was accessed via experimental studies [23–28]. PG was found to provide efficient inhibition against various cancer cell lines but was not toxic to normal cells [24,25]. The purified PG has been shown to be non-genotoxic in biological assessments used to determine mutagenic potential, including the Ames test and in vivo micronucleus [26]. A previous report also confirmed the safety of PG and its benefits to soil biodiversity at a high dose of 500 μg/mL [23]. PG is also safe when used to evaluate toxicity in eukaryotic models, such as the nematode *Caenorhabditis elegans* [27] and mice [28]. Its safety for mice is a valuable demonstration of the potential application of PG in higher organisms.

A few studies have discussed the virulence of PG in certain cases. However, the results require further scientific evidence to clarify the virulence issue. Various extracts of PG in organic solvents including kerosene ether, chloroform, acetone, ethanol, and methanol have been identified as potentially toxic to chicken embryos. A previous study warned about the choice of a conditioning solvent of suitable modulation for the application potential of PG [176]. PG-induced genomic damage was initially observed in a cell line that is normally extracted from monkey liver [177]. The combination of PG and Doxorubicin (a cancer chemotherapeutic drug) may slightly increase the cytotoxicity of Doxorubicin in tests of normal cells. However, determining the dose of PG/Doxorubicin that causes cytotoxic elongation in normal cells should be considered comprehensively [178]. Lazic et al. [120] proved that PG and its dibrominated derivatives (PG-Br and PG-Br2) did not affect *Caenorhabditis elegans* at a concentration of up to 50 μg/mL. Furthermore, the toxicity of the new bromination derivatives of PG on the zebrafish model also improved compared to parent PG [120]. Although the safety and toxicity of PG have been explored via experiments, further studies are still needed to evaluate comprehensively the toxicity of PG in each specific field.

## 4. Conclusions and Perspectives

This review summarized and updated eco-friendly substrates for fermentation to produce PG, and the benefit of PG production on a large scale (bioreactor system) was also clarified. Moreover, the potential uses of PG in agricultural applications were presented in detail. Based on the existing literature, PG has a promising use in large-scale production, even on an industrial scale. It is also considered as a potential candidate in management of multi-agents that are harmful to crops. This topic is interesting and important for the scientific community and must be further studied due to its applicability in practical agriculture in the future.

Although PG was produced using many types of byproducts/wastes, most of these materials are from fishery or agro-industrial sources. Nonetheless, there is still a need to find other byproducts in abundant amounts with nutrients, such as food processing or agricultural byproducts. To date, research related to PG production on a large scale is still very limited; hence, more studies are needed, especially on an industrial scale, to assess the exact PG productivity in industrial conditions in the future. Reports on the activities of PG in agricultural practices mostly focused on the inhibition of pathogenic fungi, whereas the tests on other harmful organisms are still very limited. Thus, subsequent studies should extend the scope of the bioactivity of this pigment to other objects, especially those that harm popular and economic crops. Furthermore, there is a need to increase greenhouse and field tests for potential bioactivities to obtain the optimum application of PG. In agricultural practices, potential bioactivities of PG in nanoscales are still not exploited, making it a promising research direction. Finally, the mechanisms of PG related to respective bioactivity in agriculture should be clarified and combined with other experimental findings to make a robust scientific basis for its application in the near future.

**Author Contributions:** Conceptualization, T.H.N., S.-L.W. and V.B.N.; writing—original T.H.N.; writing—review and editing, S.-L.W. and V.B.N.; supervision, S.-L.W. and V.B.N.; project administration, S.-L.W. All authors have read and agreed to the published version of the manuscript.

**Funding:** This research was supported in part by the Ministry of Science and Technology, Taiwan (MOST 111-2320-B-032-001; MOST 111-2923-B-032-001) and a grant from the Ministry of Education and Training, Vietnam (B2022-TTN-06).

**Data Availability Statement:** Not applicable.

**Conflicts of Interest:** The authors declare no conflict of interest.

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
