# Peer review of "Recent Advances in Eco-Friendly and Scaling-Up Bioproduction of Prodigiosin and Its Potential Applications in Agriculture"

_agronomy, doi:10.3390/agronomy12123099_

Round 1

Reviewer 1 Report

The manuscript entitled “Recent advances in eco-friendly and scaling-up bioproduction  of prodigiosin and its potential applications in agriculture” presented by Thi Hanh Nguyen and co-authors sums up recent findings for prodigiosin production and bioactivity, particularly the mediums used by now for the cultivation and impact of different compounds on the production of the pigment.

This work needs some minor changes and a few more paragraphs about toxicology studies before it can be published in the Agronomy journal.

Ln 25: in-vitro is an incorrect way of writing, it should be in vitro (correct throughout the text, example ln 515…)

Ln 26: “The mechanism of action of prodigiosin, related to its potent”

Recently (the year 2022) several publications have been published showing studies with the aim of improving the activity of the pigment by derivatization and using some eco-friendly approaches in cultivation and scale-up production, include those studies in this review.

Figure 3: the quality of this picture should be better, all the pictures are blurry and without a clean margin.

Ln.713: Gram-negative and Gram-positive (capital letter)

I can’t entirely agree with your statement made in the conclusion paragraph, stating that this review discusses for the first time large-scale production and bioactivity in detail. The bioactivity was described many times in numerous publications, together with toxicology studies. Your review lacks information about toxicology studies for this compound. Prodigiosin with all the potential as an antibacterial agent has the major disadvantage of being toxic to various cells, which was shown in in vitro studies so far.  

The text should be formatted properly

Author Response

Reviewer #1:

Comments and Suggestions for Authors:

The manuscript entitled “Recent advances in eco-friendly and scaling-up bioproduction  of prodigiosin and its potential applications in agriculture” presented by Thi Hanh Nguyen and co-authors sums up recent findings for prodigiosin production and bioactivity, particularly the mediums used by now for the cultivation and impact of different compounds on the production of the pigment.

This work needs some minor changes and a few more paragraphs about toxicology studies before it can be published in the Agronomy journal.

Reply: Thanks for your suggestion. Authors supplemented the section 3.4. The toxicology of prodigiosin in the revised version.

Ln 25: in-vitro is an incorrect way of writing, it should be in vitro (correct throughout the text, example ln 515…)

Reply: Authors checked carefully and fixed all these mistakes throughout the text.

Ln 26: “The mechanism of action of prodigiosin, related to its potent…”

Reply: This sentence was modified as “The action mechanisms related to some potential bioactivities and the toxicology studies of prodigiosin were also discussed”.

Recently (the year 2022) several publications have been published showing studies with the aim of improving the activity of the pigment by derivatization and using some eco-friendly approaches in cultivation and scale-up production, include those studies in this review.

Reply: Some reports published in 2022 have been added in this review as below. Thanks for your suggestion to improve the quality of manuscript.

Lazic J, Skaro Bogojevic S, Vojnovic S, Aleksic I, Milivojevic D, Kretzschmar M, Gulder T, Petkovic M, Nikodinovic-Runic J. Synthesis, Anticancer Potential and Comprehensive Toxicity Studies of Novel Brominated Derivatives of Bacterial Biopigment Prodigiosin from Serratia marcescens ATCC 27117. Molecules. 2022 Jun 9;27(12):3729.

Atim Asitok, Maurice Ekpenyong, Ubong Ben, Richard Antigha, Nkpa Ogarekpe, Anitha Rao, Anthony Akpan, Nsikak Benson, Joseph Essien & Sylvester Antai (2022) Stochastic modeling and meta-heuristic multivariate optimization of bioprocess conditions for co-valorization of feather and waste frying oil toward prodigiosin production, Preparative Biochemistry & Biotechnology, DOI: 10.1080/10826068.2022.2134891

 Sy Le Thanh Nguyen, Tien Cuong Nguyen, Thi Tuyen Do, Trong Luong Vu, Thi Thao Nguyen, Thi Thao Do, Thi Hien Trang Nguyen, Thanh Hoang Le, Dinh Kha Trinh, Thi Anh Tuyet Nguyen, "Study on the Anticancer Activity of Prodigiosin from Variants of Serratia Marcescens QBN VTCC 910026", BioMed Research International, vol. 2022, Article ID 4053074, 11 pages, 2022. https://doi.org/10.1155/2022/4053074

Figure 3: the quality of this picture should be better, all the pictures are blurry and without a clean margin.

Reply: The authors considered the feedback of all reviewers requiring the removal of Figure 2 and Figure 3 due to the low quality of images, and meanwhile unnecessary. Thus, the authors decided to remove these Figures.

Ln.713: Gram-negative and Gram-positive (capital letter)

Reply: Authors checked carefully and fixed all these mistakes throughout the text.

I can’t entirely agree with your statement made in the conclusion paragraph, stating that this review discusses for the first time large-scale production and bioactivity in detail. The bioactivity was described many times in numerous publications, together with toxicology studies. Your review lacks information about toxicology studies for this compound. Prodigiosin with all the potential as an antibacterial agent has the major disadvantage of being toxic to various cells, which was shown in in vitro studies so far. 

Reply: The opening paragraph was modified to be more clear and more specific. The section 3.4. The toxicology of prodigiosin was also added. The details were presented in the revised manuscript. Thanks for your comment to enhance the quality of the manuscript.

The text should be formatted properly

Reply: The manuscript was checked and formated according to requirement of journal.

Reviewer 2 Report

1. Many reviews about PG have been published

2-Abstract  need to rewrite due to repeating many sentences 

3- some references in text such as Sumathi et al., 2014 , Wei et al., 2005, Chenqiang et al., 2019,Shahitha et al., 2012, Nguyen et al., 2020,Tran et 190 al., 2021,  Nguyen et al., 2022, Nisarg et al., 2021, Ozdal et al., 212 2015,Ahmed et al., 2012 must be changed NO [ ]

4. line 48 , 49 :  the same references  repaeated :  culture conditions, 47 additives, and medium compositions [2,3,10,12,13]. Besides, current reviews have until 48 now discussed the bioactivities of PG in medicine [2,3,10,13,14].

 5- some figures not requried suuch as fig. 2

6. Discuss the following : Qadri et al., 1973 discovered the role of methionine in PG biosynthesis and 298 observed that the addition of this amino acid shortened the delay time in the PG 299 biosynthesis process, to thus increase its productivity

7. add another references at the end :  In nature, microbes exist in a complex population, and their interaction is also a 329 factor to be considered for enhancing the production of secondary metabolites [76]. such as 

Abdel Ghany TM (2014) Eco-friendly and Safe Role of Juniperus procera in Controlling of Fungal Growth and Secondary Metabolites. J Plant Pathol Microbiol 5: 231. doi:10.4172/2157-7471.1000231

Abdelghany, T.M., El-Naggar, M.A., Ganash, M.A. et al. PCR Identification of Aspergillus niger with Using Natural Additives for Controlling and Detection of Malformins and Maltoryzine Production by HPLC. BioNanoSci. 7, 588–596 (2017). https://doi.org/10.1007/s12668-017-0455-6 

8. Saccharomyces Cerevisiae  changed to Saccharomyces cerevisiae

9. line 463 what this:  - a

10 fig. 3  please delete

11. title Table 5. Anti-fungal activity of prodigiosin for agriculture must change to :Table 5. Anti-fungal activity of prodigiosin for agriculture applications

12 IC50 must change to IC50

13. IC50 - Concentration to inhibit 50% pleses delete the identification of IC50  from table use only the IC50

14. line 661 Pythium ultimum fungi or Pythium ultimum fungus

15.  line 704 please add another references at end  : Currently, green synthesis by nanotechnology combined with natural ac- 704 tive ingredients is a new trend to improve the formulations of pesticides in terms of effec- 705 tiveness, safety, and environmental protection [146]. such as: Abdelghany, T.M., Al-Rajhi, A.M.H., Al Abboud, M.A. et al. Recent Advances in Green Synthesis of Silver Nanoparticles and Their Applications: About Future Directions. A Review. BioNanoSci. 8, 5–16 (2018). https://doi.org/10.1007/s12668-017-0413-3

Hasanin, M., Al Abboud, M.A., Alawlaqi, M.M. et al. Ecofriendly Synthesis of Biosynthesized Copper Nanoparticles with Starch-Based Nanocomposite: Antimicrobial, Antioxidant, and Anticancer Activities. Biol Trace Elem Res 200, 2099–2112 (2022). https://doi.org/10.1007/s12011-021-02812-0

Amr H. Hashem,Mohamed A. Al Abboud,Mohamed M. Alawlaqi,Tarek M. Abdelghany,Mohamed Hasanin

 2021

Synthesis of Nanocapsules Based on Biosynthesized Nickel Nanoparticles and Potato Starch: Antimicrobial, Antioxidant, and Anticancer Activitydoi.org/10.1002/star.202100165

Abdelghany, T.M., Al-Rajhi, A.M.H., Almuhayawi, M.S. et al. Green fabrication of nanocomposite doped with selenium nanoparticle–based starch and glycogen with its therapeutic activity: antimicrobial, antioxidant, and anti-inflammatory in vitro. Biomass Conv. Bioref. (2022). https://doi.org/10.1007/s13399-022-03257-8

16. avoide similarity between abstract and conclusion

Author Response

Reviewer #2:

  1. Many reviews about PG have been published

Reply: Some parts in the introduction were modified for emphasising the novel points of this review work comparing to the previous review articles. Thanks for your comments.

2-Abstract need to rewrite due to repeating many sentences

Reply: The abstract section was checked and revised to avoid sentence repetition errors.

3- some references in text such as Sumathi et al., 2014 , Wei et al., 2005, Chenqiang et al., 2019,Shahitha et al., 2012, Nguyen et al., 2020,Tran et 190 al., 2021,  Nguyen et al., 2022, Nisarg et al., 2021, Ozdal et al., 212 2015,Ahmed et al., 2012 must be changed NO [ ]

Reply: Some references were changed to No. [] for suitable.

  1. line 48 , 49 : the same references repaeated :  culture conditions, 47 additives, and medium compositions [2,3,10,12,13]. Besides, current reviews have until 48 now discussed the bioactivities of PG in medicine [2,3,10,13,14].

Reply: This mistake was revised. Thanks for your reminding.

5- some figures not requried suuch as fig. 2

Reply: The authors considered the feedback of all reviewers requiring the removal of Figure 2 and Figure 3 due to unnecessary and low quality of images. The authors decided to delete them to improve the manuscript's quality.

  1. Discuss the following: Qadri et al., 1973 discovered the role of methionine in PG biosynthesis and 298 observed that the addition of this amino acid shortened the delay time in the PG 299 biosynthesis process, to thus increase its productivity

Reply: The authors have discussed some relevant information in the attached manuscript.

  1. add another references at the end: In nature, microbes exist in a complex population, and their interaction is also a 329 factor to be considered for enhancing the production of secondary metabolites [76]. such as

Abdel Ghany TM (2014) Eco-friendly and Safe Role of Juniperus procera in Controlling of Fungal Growth and Secondary Metabolites. J Plant Pathol Microbiol 5: 231. doi:10.4172/2157-7471.1000231

Abdelghany, T.M., El-Naggar, M.A., Ganash, M.A. et al. PCR Identification of Aspergillus niger with Using Natural Additives for Controlling and Detection of Malformins and Maltoryzine Production by HPLC. BioNanoSci. 7, 588–596 (2017). https://doi.org/10.1007/s12668-017-0455-6

Reply: These references were added per your suggestion.

  1. Saccharomyces Cerevisiae changed to Saccharomyces cerevisiae

Reply: This mistake was revised. Thanks for your finding.

  1. line 463 what this: - a

Reply: It’s “a kind byproduct”. This mistake was checked and revised.

10 fig. 3 please delete

Reply: The authors have considered and removed it according to your suggestion.

  1. title Table 5. Anti-fungal activity of prodigiosin for agriculture must change to : Table 5. Anti-fungal activity of prodigiosin for agriculture applications

Reply: The title Table 5 was fixed as your suggestion.

12 IC50 must change to IC50

Reply: The authors changed all text.

  1. IC50 - Concentration to inhibit 50% pleses delete the identification of IC50 from table use only the IC50

Reply: the identification of IC50 in the table was removed and noted under table.

  1. line 661 Pythium ultimum fungi or Pythium ultimum fungus

Reply: It was edited into fungus.

  1. line 704 please add another references at end: Currently, green synthesis by nanotechnology combined with natural ac- 704 tive ingredients is a new trend to improve the formulations of pesticides in terms of effec- 705 tiveness, safety, and environmental protection [146]. such as: Abdelghany, T.M., Al-Rajhi, A.M.H., Al Abboud, M.A. et al. Recent Advances in Green Synthesis of Silver Nanoparticles and Their Applications: About Future Directions. A Review. BioNanoSci. 8, 5–16 (2018). https://doi.org/10.1007/s12668-017-0413-3

Hasanin, M., Al Abboud, M.A., Alawlaqi, M.M. et al. Ecofriendly Synthesis of Biosynthesized Copper Nanoparticles with Starch-Based Nanocomposite: Antimicrobial, Antioxidant, and Anticancer Activities. Biol Trace Elem Res 200, 2099–2112 (2022). https://doi.org/10.1007/s12011-021-02812-0

Amr H. Hashem,Mohamed A. Al Abboud,Mohamed M. Alawlaqi,Tarek M. Abdelghany,Mohamed Hasanin 2021. Synthesis of Nanocapsules Based on Biosynthesized Nickel Nanoparticles and Potato Starch: Antimicrobial, Antioxidant, and Anticancer Activity. doi.org/10.1002/star.202100165

Abdelghany, T.M., Al-Rajhi, A.M.H., Almuhayawi, M.S. et al. Green fabrication of nanocomposite doped with selenium nanoparticle–based starch and glycogen with its therapeutic activity: antimicrobial, antioxidant, and anti-inflammatory in vitro. Biomass Conv. Bioref. (2022). https://doi.org/10.1007/s13399-022-03257-8

Reply: These references were added in the revised version. Thanks for your suggestion these reports for enrichment of information.

  1. avoide similarity between abstract and conclusion

Reply: Thanks for your mention. The abstract part and conclusion part were modified for more clear and specific.

Reviewer 3 Report

This review article can be accepted for publication in Agronomy-MDPI after taking my following comments into consideration:

1.  Several parts of this review article need revision, especially from a linguistic point of view

2. Researchers have to work according to the journal’s instructions regarding the names of scientists and references, the review should be done accordingly in all parts of the review article.

3.     The topic of this work is interesting and important to the scientific community and needs further study due to the possibility of its practical and practical use in agriculture.

4.    The introduction part contains a lot of redundant and unnecessary

5.   Researchers must respond to my comments included in the attached file

6.  The references should be written with paying attention to the recent ones in the topic of the current manuscript

 Detailed comments on the manuscript are in the attached file

Author Response

Reviewer #3:

This review article can be accepted for publication in Agronomy-MDPI after taking my following comments into consideration:

  1. Several parts of this review article need revision, especially from a linguistic point of view

Reply: Some parts including the abstract, introduction, and conclusion were revised as your comment and other review comments for more scientific and clear. The modified contents were presented in the revised manuscript. Thanks for your comment to improve the quality of the manuscript.

  1. Researchers have to work according to the journal’s instructions regarding the names of scientists and references, the review should be done accordingly in all parts of the review article.

Reply: All these mistakes was revised in the text. Thanks for your reminder.

  1. The topic of this work is interesting and important to the scientific community and needs further study due to the possibility of its practical and practical use in agriculture.

Reply: Thanks for your positive comments and having interesting view on our work. This comment is necessary and confirms the need for further research on PG and the value of PG in practical agriculture in the future. We also added this information in the revised manuscript.

  1. The introduction part contains a lot of redundant and unnecessary

Reply: The introduction was modified for clearer , and the detail was presented in the revised manuscript.

  1. Researchers must respond to my comments included in the attached file

- Keywords: “prodigiosin” this word are presented in the title , you can change it by another one.

Reply: We changed it to “red pigment”.

- In lines 35-42, the authors mentioned several species not this species only.

 In the abstract, you mentioned this species only, please explain.

Reply: Prodigiosin was produced from various bacterial strains. Among them, mainly source producing PG is Serratia marcescens. Thus, in the abstract part, the authors only mention this main strain. For better understanding, the abstract was revised as suitable accordingly your comment.

- Line 32-33, add relevant reference.

Reply: The authors were added relevant reference as your suggestion.

- This sentence (line 49,50) need more explanation

Line 52, not correct

Reply: The paragraph containing these sentences was revised and the detail in the attached revised version.

- Line 74-75, add relevant reference.

Reply: The authors were added relevant reference as your suggestion.

- C/N source meaning

Reply: It means Carbon/Nitrogen source. The authors edited it.

- Line 110-111, this part is more suitable for the materials and methods section

Reply: Due to in this review mentioned about two kinds of fermentation in PG production including Liquid-state fermentation (LSF) and Solid-state fermentation (SSF). We only showed a bit of defining them for more understanding of these types of fermentation. In addition, this is review paper, as such, we suggested remain this information in this part.

- Change mg/L to mg/L-1 and similarly mistakes about unit

Reply: The authors checked and edited them in the text.

- Delete sentence “Microbial fermentation is the primary source of PG production.”

Reply: This sentence was removed accordingly the suggestion.

- Line 184-185, add relevant reference.

Reply: The authors were added relevant reference as your suggestion.

- Line 251-253, shorten sentences

Reply: This sentences were modified shorten.

- Line 277-281, shorten sentences and add relevant reference.

Reply: This sentences were modified shorten and supple relevant reference.

- Line 414-415, add relevant reference.

Reply: The authors were added relevant reference as your suggestion.

- Line 464-466, add relevant reference.

Reply: This is a comment based on the results summarized in Table 4. Thus, the authors also added a note as "based on the results synthesized in table 4" following this sentence for easier understanding.

- Line 486-492, No require to these sentences, do not repetion remove these sentences and start directicly by the potential application of PG in Agriculture

Reply: This paragraph was modified for suitable. Thanks for your comment.

- Figure 3. You can reduce the size of an image to improve its quality

Reply: The authors considered the feedback of all reviewers requiring the removal of Figure 2 and Figure 3 due to unnecessary and low quality. The authors decided to delete them to improve the manuscript's quality.

- Line 517-519, delete, Be specific with focucing on the topic only

Reply: This paragraph was modified for suitable. Thanks for your comment to improve the quality of manuscript.

- Line 591-592, delete

Reply: We were removed it as your suggestion.

- Is there evidence or previous references to support this information? “Besides the potential activity in in vitro, PG also showed promising impacts when applied in in-vivo, greenhouse, or field conditions. However, until now, not much re-search has assessed the bioactivities of this pigment on a larger scale, especially in greenhouse conditions or fields”

Reply: This is an author's comment based on reviewing literatures summarised in sections 3.1 and 3.2. In fact that reports about bio-activity in the agriculture of PG have mainly evaluated in vitro conditions, with only some reports conducted in the greenhouse and rarely study tests in the field.

- 4. Conclusions and Perspectives. In this part, a conclusion must be written in a clear and specific sentence that leads to a specific meaning, and the future outlook towards the use of this dye in application in agriculture must be mentioned, according to what was mentioned in the title of the manuscript.

Reply: The opening paragraph was modified to be more clear and more specific as your suggestion. The details were presented in the revised manuscript. Thanks for your comment to enhance the quality of the manuscript.

  1. The references should be written with paying attention to the recent ones in the topic of the current manuscript

Reply: Some recent reports published have been added revised manuscript. Thanks for your suggestion to improve the quality of manuscript.

Round 2

Reviewer 2 Report

good